# Patient General Condition at Diagnosis: A Systematic Evaluation for Adults Diagnosed with Hematologic Malignancies

**DOI:** 10.3390/jpm10030106

**Published:** 2020-08-27

**Authors:** Fernando Ramos, Paola González-Carmona, María Isabel Porras-Guerra, Sonia Jiménez-Mola, Ana María Martínez-Peláez, Agustín Blanco-Cabielles, Saray Conde, Abdolah Ahmadi, Marta Castellanos, Seila Cerdá, Natalia de las Heras, Elisa Menéndez, Fernando Escalante, Silvia Fernández-Ferrero, Tamara Lado, Violeta Martínez-Robles, Filomeno Rondón, Irene Padilla, María Jesús Vidal, María Lavinia Villalobos, Saad Yacoubi, Francisco Javier Idoate-Gil, José Antonio Rodríguez-García

**Affiliations:** 1Department of Hematology, Hospital Universitario de León, 24008 León, Spain; paola.g.c@outlook.es (P.G.-C.); abdolah@saludcastillayleon.es (A.A.); mcastellanosa@saludcastillayleon.es (M.C.); seila.cr@gmail.com (S.C.); nherasr@saludcastillayleon.es (N.d.l.H.); fescalanteb@saludcastillayleon.es (F.E.); smfernandez@saludcastillayleon.es (S.F.-F.); tamara.lado@gmail.com (T.L.); viomartinez@saludcastillayleon.es (V.M.-R.); frondon@saludcastillayleon.es (F.R.); irene.padilla.conejo@hotmail.es (I.P.); mjvidalman@saludcastillayleon.es (M.J.V.); mlvillalobos@saludcastillayleon.es (M.L.V.); y_saad22_7@hotmail.com (S.Y.); jrodriguezgar@saludcastillayleon.es (J.A.R.-G.); 2Department of Geriatrics, Hospital Universitario de León, 24008 León, Spain; miporras@saludcastillayleon.es (M.I.P.-G.); soniajimenezmola@hotmail.com (S.J.-M.); fidoate@saludcastillayleon.es (F.J.I.-G.); 3Department of Nursing, Hospital Universitario de León, 24008 León, Spain; anamartinez-pelaez@hotmail.com (A.M.M.-P.); andydue22@hotmail.com (A.B.-C.); sarayconde35@gmail.com (S.C.); paulaeely@hotmail.com (E.M.)

**Keywords:** personalized medicine, hematologic neoplasms, aged, Lee Index, Geriatric Assessment in Hematology, feasibility, agreement

## Abstract

Several societies have published recommendations for evaluating older adults with cancer in standard conditions. It is vital to assure a proper systematic patient condition evaluation, not only in the oldest (geriatric assessment) but in all adult patients. We have investigated the feasibility of a systematic evaluation of the general condition of all patients diagnosed with hematologic malignancies, and the degree of acceptance by the clinical team, in a prospective cohort of 182 consecutive adults, by using the ECOG performance status scale (ECOG, age 18 and over, 18+), Lee Index for Older Adults (LEE, 50+), Geriatric Assessment in Hematology (GAH, 65+), and the Comprehensive Geriatric Assessment (CGA, 75+). Clinical team acceptance was analyzed with a visual analogue scale, and the objective feasibility was calculated as the proportion of patients that could be finally evaluated with each tool. Acceptance was high, but the objective feasibility was progressively lower as the complexity of the different tools increased (ECOG 100%, LEE 99.4%, GAH 93.2%, and CGA 67.9%). LEE and GAH categories showed a weak concordance (Cohen’s Kappa 0.24) that was slight between LEE and CGA (Kappa 0.18). Unexpectedly, we found no significant association between the GAH and CGA categories (*p* = 0.16). We confirm that a systematic evaluation of all adult patients diagnosed with hematologic malignancies is feasible in daily practice by using an age-adapted approach. Direct comparisons among the different predictive tools in regard to patients’ tolerance to treatments of different intensities must be a priority research subject in the coming years.

## 1. Introduction

Patient condition (PC) evaluation has a key relevance for the treatment of patients diagnosed with malignancies. It is generally performed in a non-standardized manner by the attending physician, followed by the assignation of ECOG performance status (ECOG) and, ideally, a comprehensive geriatric assessment (CGA) in the oldest patients. At the present time, it is vital to implement a systematic PC evaluation, not only in the oldest [1] but in all adult patients.

ECOG depends on both prior condition and the new disease, and may show short-term variations after *pre-phase* therapeutic interventions. Prior condition is a determinant of the expected survival and can be estimated by the Lee Index for Older Adults (LEE) [2] in those aged 50 and above (50+). This index can be calculated online at https://eprognosis.ucsf.edu/lee.php. It has also been shown that LEE is an independent prognosticator of overall survival in myelodysplastic syndromes [3] and probably in chronic myelomonocytic leukemia and acute myeloid leukemia [4].

Older adults diagnosed with cancer must be assessed [5,6,7,8,9] by a geriatric team or a “geriatric trained physician”, upfront or after a screening procedure. Nevertheless, many hospitals lack optimal geriatric facilities and the Spanish Group for Geriatrics in Hematology has developed an abridged geriatric scale (Geriatric Assessment in Hematology, GAH) to simplify this process [10,11]. GAH is now on an advanced stage of development and validation, and it is owned by Celgene S.L.U. who has developed an online calculator (https://gah.proyectocelvision.net/) to spread its clinical use.

## 2. Methods

### 2.1. Study Aim and Cohort Description

The aim of this study was to investigate the feasibility of a systematic and age-adapted evaluation of the PC of all patients diagnosed with hematologic malignancies (HMs) by means of dedicated clinical tools, as well as its acceptance by the clinical team. We intended to analyze prospectively the general condition of all patients diagnosed with hematologic malignancies in the Unit of Clinical Hematology, Department of Hematology and Blood Transfusion at Hospital Universitario de León during the calendar year 2018. The study was conducted after approval by the Institutional Ethics Committee and the procedures followed were in accordance with the Helsinki Declaration of 1975, as revised in Fortaleza (Brasil) in 2013. All the patients analyzed gave prior written informed consent.

### 2.2. Measures

The ECOG scale was to be measured in all patients, LEE in patients 50 and over (50+), GAH in 65+, and CGA in 75+. We selected the age application brackets for the different tests following the design of the original manuscripts describing their use. In regard to CGA, although it can be used from age 65, our geriatric colleagues considered that 75 years is the most appropriate cutoff to use CGA as frailty under that age is very uncommon [12].

Each scale was applied once to each patient by one professional. The ECOG performance status and Lee scale for older adults were measured by the attending hematologists (either licensed or in training), GAH scale was measured by the ward/office hematology nurses that had been trained on its use, and CGA was performed by the geriatricians. Additional details on the clinical team characteristics can be found in Appendix A.

The degree of acceptance of each professional with the general strategy and the different tools were checked by completing an ad hoc visual analogue scale consisting of a question on the personal opinion of the investigator on the general strategy as well as on each individual tool, that had to be answered by drawing a mark on a 100 mm horizontal line labeled 0 at the beginning and 100 at its end. Objective feasibility was calculated as the proportion of patients that actually completed each tool. We describe the median and quartiles of the distribution of each covariate.

### 2.3. Statistical Analysis

Lastly, we calculated the statistical association (Chi-square, exact method) and agreement (Cohen’s Kappa) between the different tools and the clinical characteristics of the participants (both patients and health providers). For this purpose, dichotomization of covariates was performed according to the common clinical practice (ECOG, 0–2 vs. other; CGA, robust vs. other), published cutoffs (GAH, score up to 42 vs. 43 or more), or the median of the distribution (LEE, score less than 6 vs. score 6 and over). All signification tests were two-sided and a *p*-value < 0.05 was considered statistically significant. We did not correct nominal *p*-values for multiple comparisons.

## 3. Results

### 3.1. Characteristics of the Participants

Two hundred and nine patients were initially considered and twenty-seven (12.9%) were finally excluded (diagnosis out of the time frame 2, duplicates 3, consent denial 2, lack of documental demonstration of consent 20).

We analyzed 182 patients (99 males, 83 females), aged 19–97 years (median 72, p25–p75: 61–82), diagnosed with acute leukemias (22), myeloproliferative neoplasms (20), myelodysplastic syndromes (22), myeloproliferative/myelodysplastic neoplasms (5), Hodgkin’s lymphoma (6), other lymphomas (60), chronic lymphocytic leukemias and allied neoplasms (20), and multiple myeloma (27). The subsets of patients that were candidates to each scale were: ECOG 182 (18+), LEE 163 (50+), GAH 117 (65+), CGA 84 (75+).

Eight patients aged 65–74 and 27 aged 75 and older did not have their patient condition analyzed for different reasons. Patient distribution according to the different scales categories, as well as the reasons why 27 out of 84 patients aged 75 and over did not receive the CGA are shown in the Appendix A.

### 3.2. Tool Applicability and Clinical Correlation

The ECOG scale could be administered to every participant (182/182, 100%), LEE scale to 162/163 (99.4%), GAH scale to 109/117 (93.2%), and CGA to 57/84 (67.9%). Interestingly, GAH and CGA categories were associated to patients’ diagnostic group (*p* < 0.03 and *p* < 0.01, respectively), but this was not the case for ECOG and LEE (*p* = 0.13 and *p* = 0.27, respectively, Appendix A). In this regard, frequency of high-score GAH (43 or more score points) was minimal in Hodgkin’s lymphoma and myeloproliferative neoplasms (0% and 12.5%, respectively), and maximal in multiple myeloma, acute leukemia, and other lymphomas (76.5%, 66.3%, and 55.6%, respectively), while frequency of CGA non-robust was minimal in chronic lymphocytic leukemia and related disorders (15.4%) as well as in myelodysplastic syndromes (23.1%) and maximal in lymphomas (100% in Hodgkin’s and 72.2% in the other lymphomas (Appendix A)).

Median acceptance score of the global strategy (i.e., the intention to evaluate the patient’s condition by means of dedicated tools in lieu of just the clinical eye) by the research team, in a scale of 0–100, was 78 (p25–p75: 72–93) and that of the individual tools was 81 for ECOG (84–95), 78 for LEE (69–100), 81 for GAH (72–93), and 80 for CGA (69–94). We did not observe any statistically significant difference according to team members age, gender, professional level, degree of specialization, functional role, or degree of implication in the study (data on file).

### 3.3. Tool Association and Agreement

We observed a statistically significant association (*p* < 0.01) and a weak agreement (Kappa 0.24) between LEE and GAH categories in the 109 patients aged 65 and over, that had both tools measured (Table 1A). In those aged 75+, LEE categories showed a statistically significant association (*p* < 0.02) as well as a slight agreement (Kappa 0.18) with CGA categories (*n* = 57, Table 1B). By contrast, we did not find any significant association between GAH and CGA categories (*n* = 53, *p* = 0.16).

In regard to the ECOG performance status, we found a statistically significant association between ECOG and GAH (*p* < 0.01, Kappa 0.21) or CGA categories (*p* < 0.01, Kappa 0.28), but not with LEE categories (*p* = 0.34).

## 4. Discussion

PC is a determinant of clinical evolution and may be influential in patient’s attitude to the disease, that has been shown to be determinant of quality-of-life in patients diagnosed with some hematologic neoplasms [13]. Its evaluation is important for patient care and contributes to personalized medicine, that must not rest solely on the side of disease but also on the patient’s one. The demographic evolution of most industrialized countries calls for an age-sensitive approach and the present coronavirus pandemic crisis makes this extremely important. The clinical relevance of CGA in the oldest adults without cancer is beyond any doubt, has also been recommended in cancer patients by several scientific societies [6,7,8], and it is being performed in patients diagnosed with HMs [9,14,15]. Nevertheless, gender, smoking habit, nutrition, comorbidity, etc., are not exclusive of the oldest adults and are also relevant for those in their fifties, while their impact is probably limited in younger patients. Thus, an age-adapted strategy may assist the attending team to tailor its activities to the likelihood that the patient suffers from age-related derangements, as well as to reduce variability in clinical practice and the subsequent discrepancies within/between the attending team/s. To our knowledge, this study is the first one to address the feasibility of a systematic PC evaluation in the wide range of adult age in daily malignant hematology practice.

According to its original design, only the unidimensional ECOG performance status scale can be applied to the full spectrum of adult patients with HMs. In the elderly (65 and over) the multidimensional CGA is the gold standard, but other strategies are possible, such as a sequential approach (a screening tool followed by a full CGA for those qualifying), an abridged CGA (such as GAH) or modified CGA strategies that include just a few domains [16]. Some of them have been or are being studied in a few therapeutic scenarios, but have not yet been tested in the wide range of HMs. Finally, the multidimensional Lee Index for Older Adults covers the orphan 50–64 age bracket (actually, it can be applied to patients aged 50 and older), and is a clear opportunity for a quick PC categorization in patients with HMs.

As expected, and despite the proactive attitude of our research team, the objective feasibility of each tool is a function of its complexity, ECOG4 and LEE being the most feasible, and maybe the most efficient in periods of crisis. In addition, we have found that both GAH and CGA seem to be influenced by patients’ diagnosis and ECOG, while LEE does not, which makes LEE less dependent on the new disease.

In concordance with other recently published data from multiple myeloma patients evaluated with different strategies to study frailty [17,18], we have found that the agreement among the different tools to evaluate PC is far from perfect in patients 65+ and, consequently, they are not readily interchangeable. This is not an unexpected finding since the various tools address different domains (Table 2) and have different goals (LEE, life-expectancy estimation; GAH, plan B for geriatric assessment if CGA is not available; CGA, optimal geriatric evaluation). We hypothesized that some degree of agreement between the categories of the different scales was to be expected since they share one or more domains, but the low observed agreement among them makes the tool choice very relevant for patient allocation to the very PC categories. The hematologists should be conscious of each tool’s scope and age-applicability and use the best endorsed and less complex tool for each age bracket. Direct comparisons among the different predictive tools, regarding patients’ tolerance to treatments of different intensities must be a priority research subject in the coming years, because we have shown that tool feasibility is inversely related to its complexity, and CARG [19] and CRASH [20] are not readily applicable to patients with HMs.

Our work has some limitations. First, twenty-seven patients were excluded for different reasons and this might have caused a selection bias. Nevertheless, the demographical and clinical characteristics of the excluded patients did not differ from those analyzed except for the absence of patients younger than 50. Second, we have not analyzed the intra- or inter-observer agreement since each scale was applied only once by a single investigator, and they should also be addressed in future investigations.

Although we have performed several scales in the same individual for research purposes with the intention to analyze their correlation and agreement, we propose the following algorithm (Figure 1): First, to evaluate the impact of the new disease in all patients, regardless of age, by means of the ECOG performance status, as it is a very relevant prognostic factor for patients with HMs and a critical selection criterion for most clinical trials; second, to evaluate patient life-expectancy prior to the development of the new disease in 50+ by means of the Lee Index, in order to estimate the potential life-years lost if the patient is not treated by a curative intent; and finally, whenever possible, to perform a geriatric assessment in 65+ to evaluate the eventual tolerance to standard treatment, as well as to detect any derangement that may be amenable to geriatric intervention.

Given the lack of widespread geriatric facilities in many countries, GAH might suffice initially for those patients aged 60–75. If the GAH score is higher than 42 or the patient is aged 75 and over, CGA would be mandatory, if available. In the case of disagreement among the different scales, we suggest using the most complex one (i.e., CGA > GAH > LEE > ECOG). By contrast, when facing a global health crisis such as the current COVID-19, the sequential application of just ECOG and LEE (in 50+) would be the most appropriate approach, as we think that this strategy can be easily adopted by a number of clinical teams in the worst conditions. Although our algorithm has been developed for patients diagnosed with HMs, it might also be suitable for patients with other hematologic and non-hematologic diseases, given the general purpose of the tools considered.

In conclusion, the patient condition can be assessed systematically in patients diagnosed with hematologic neoplasms by using an age-stratified strategy. Direct comparisons among the different predictive tools in regards to patients’ tolerance to treatments of different intensities must be a priority research subject in the coming years.

## Figures and Tables

**Figure 1 jpm-10-00106-f001:**
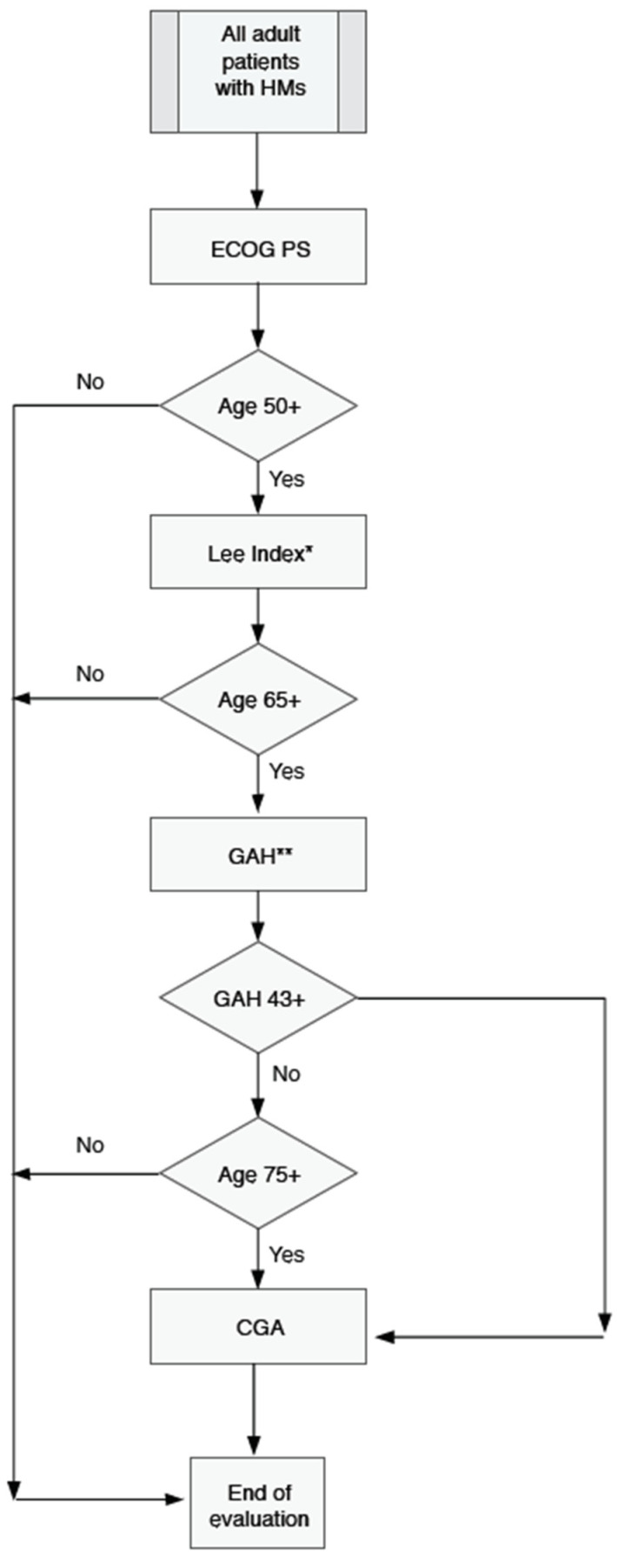
Proposed algorithm for systematic age-adapted patient general condition evaluation at diagnosis in adult patients diagnosed with hematological malignancies. HMs, Hematologic malignancies; ECOG PS, Performance status according to ECOG scale; LEE INDEX (Lowcase); Lee’s Index for Older Adults [2]; GAH, Geriatric Assessment in Hematology [10]; CGA, Comprehensive Geriatric Assessment. * In times of crisis, the evaluation might be simplified to the first two steps: ECOG and LEE in 50+. ** For patients 65+, GAH might also be substituted by the six-item version [21] of the G8 questionnaire (abnormal geriatric assessment if the score is 6 or more points) in both hematological and non-hematological patients.

**Table 1 jpm-10-00106-t001:** Associations amongst the different tools used to evaluate patient condition.

**A**						
	**ECOG**		**LEE**		**GAH**	
CGA	0–2	3–4	<6	6+	<43	43+
Robust	33	0	7	26	18	14
Other	18	6	0	24	7	14
Total	57		57		53	
*p*-value	<0.01		<0.02		0.16	
**B**						
	**GAH**					
LEE	score <43		score 43+		Total	
Score <6	24		11		35	
Score 6+	30		44		74	
Total	54		55		109	
*p*-value	<0.01					

Panel A. Association between LEE and GAH scores in patients 65+. Panel B. Association between CGA and the other scores in patients 75+. ECOG: Performance status according to the ECOG scale; LEE: Lee’s Index for Older Adults [2]; GAH: Geriatric Assessment in Hematology [10]; CGA: Comprehensive Geriatric Assessment.

**Table 2 jpm-10-00106-t002:** Dimensions covered by each tool and clinical applicability.

DIMENSIONS	ECOG	LEE	GAH	CGA
Physical performance	+	+	+	+
ADL	−	+	+	+
Comorbidity	−	+	+	+
Cognition and Mood	−	+	+	+
Nutritional status	−	+	+	+
Polypharmacy and medication review	−	−	+	+
Socio-economical issues	−	−	−	+
Subjective health status	−	−	+	+
Age	−	+	−	−
Gender	−	+	−	−
Life style	−	+	−	−
APPLICABILITY				
Age bracket	18+	50+	65+	70–75+

ECOG: Performance status according to ECOG scale; LEE: Lee’s Index for Older Adults [2]; GAH: Geriatric Assessment in Hematology [10]; CGA: Comprehensive Geriatric Assessment; ADL: Basic Activities of Daily Life. Several dimensions are measured in LEE by a single item or just a few.

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
