# Peer review of "Patient General Condition at Diagnosis: A Systematic Evaluation for Adults Diagnosed with Hematologic Malignancies"

_jpm, 2020, doi:10.3390/jpm10030106_

Round 1

Reviewer 1 Report

The authors present interesting report on the utility of various geriatric assessment for patients with hematological malignancies.

It would be helpful if the authors better described why the set certain age limits for the various test, and CGA has been used in patients who are younger the 75, and how that might affect their results.

Lastly, I wonder if the authors had any thoughts regarding using a modified CGA which has been studied in various hematological malignancies and can assess patient fitness.  

Reviewer 2 Report

This manuscript aims to identify the feasibility and acceptability of various patient condition (PC) evaluations diagnosed in the inpatient setting with a hematologic maligancy (HM) at a single institution in 2018. The methods are straightforward but lack definitions. The results are clear but diverge from the stated aims. Composition of the discussion can be significantly improved.

Major points:

  1. Please provide additional information about the clinical team (size, experience, familiarity with tested assessments), preferably in a table
  2. Please explain or include the described “ad hoc” visual analogue scale
  3. Please describe further the associations between GAH and CGA administration and diagnostic group
  4. More information about cases where application of PC tools failed is needed
  5. The proposed PC evaluation algorithm does not appear to be based on the included results as it, for instance, does not account for the “diagnostic group” that is associated with tool applicability

Minor points:

  1. Define “diagnostic group” P3 line 117
  2. Define “global strategy” P4 line 119
  3. Please comment on the lower quartile range of 4 for ECOG, which seems out of place (P4 Line 120)
  4. The relevance of Section 3.3 is unclear – do the authors hypothesize agreement across scoring systems?
  5. A deeper description of the strengths and limitations of validated PC tools as they pertain to patients with HM’s across age groups would strengthen the discussion

Reviewer 3 Report

Very nice and accurate article on performance status in haematological elderly patients. I will suggest to add a sentence in the discussion mentioning also the relevance of quality of life for haematological patients.

You might referr to

Quality of life in elderly patients with essential thrombocythaemia. An Italian multicentre study. Oliva EN, et al .Ann Hematol. 2012 Apr;91(4):527-32

Round 2

Reviewer 2 Report

I agree the manuscript has been significantly improved and could be considered for publication in JPM.